# Post-hoc Utterance Refining Method by Entity Mining for Faithful Knowledge Grounded Conversations

**Yoonna Jang**[*], **Suhyune Son**[*], **Jeongwoo Lee**[*], **Junyoung Son, Yuna Hur,**
**Jungwoo Lim, Hyeonseok Moon, Kisu Yang,** and **Heuiseok Lim**[†]
Department of Computer Science and Engineering, Korea University
{morelychee, ssh5131, time79779, s0ny, yj72722,
wjddn803, glee889, willow4, limhseok}@korea.ac.kr

## Abstract

Despite the striking advances in recent language generation performance, model-generated responses have suffered from the chronic problem of hallucinations that are either untrue or unfaithful to a given source. Especially in the task of knowledge grounded conversation, the models are required to generate informative responses, but hallucinated utterances lead to miscommunication. In particular, entity-level hallucination that causes critical misinformation and undesirable conversation is one of the major concerns. To address this issue, we propose a post-hoc refinement method called **REM**. It aims to enhance the quality and faithfulness of hallucinated utterances by refining them based on the source knowledge. If the generated utterance has a low source-faithfulness score with the given knowledge, REM mines the key entities in the knowledge and implicitly uses them for refining the utterances. We verify that our method reduces entity hallucination in the utterance. Also, we show the adaptability and efficacy of REM with extensive experiments and generative results. Our code is available at https://github.com/YOONNAJANG/REM.

## 1 Introduction

The knowledge grounded conversation (KGC; also called knowledge grounded conversation, KGD), which is a subfield of the dialogue systems, is a task of generating human-like utterances by referring to specialized knowledge such as Wikipedia[1] (Zhao et al., 2022; Li et al., 2022c). The KGC task requires the ability to generate fluent and informative utterances based on source knowledge. Considering this ability, there has been numerous concurrent research on the KGC to build models that can have conversations with human-like expertise (Ma et al.,

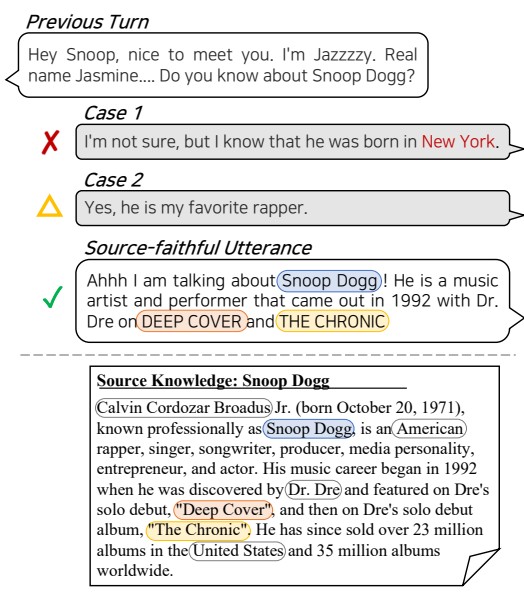

Figure 1: An illustrated example of the utterances with entity-level hallucination (*i.e.*, *case 1* and *2*) and desirable utterances in knowledge grounded conversation. In the source-faithful utterance, entities of the given source knowledge are included, making the utterance informative.

2020; Yu et al., 2022; Shuster et al., 2022a; Liu et al., 2022; Sun et al., 2023; Xiao et al., 2023).

A colossal number of training corpora and parameters have brought the recent explosion of language generation performance in pre-trained language models (PLMs) (Brown et al., 2020; Raffel et al., 2020) and large language models (LLMs) (Thoppilan et al., 2022; Touvron et al., 2023; OpenAI, 2023). However, hallucination, which has been a chronic problem in natural language generation, is still cited as one of the biggest challenges remaining unsolved (Ji et al., 2023). In the KGC task, even though the ground truth knowledge is given, the models make the error of generating hallucinated utterances not faithful to the source knowledge (Li et al., 2022a; Ji et al., 2023).

Especially, *entity-level hallucination*, generating

---

[*]These authors contributed equally to this work.
[†]Corresponding author.
[1]https://www.wikipedia.org/

names of entities that are incorrect or not present in the source document (Nan et al., 2021), causes critical misinformation and jeopardizes the flow of the conversation (Das et al., 2023). As shown in *Case 1* of Figure 1, the previous KGC models generate the utterance containing information about the wrong entity, which is not given in the knowledge. Further, the generated utterance is excessively general while not considering sufficient entities that align with the context of the conversation, as shown in *Case 2*. These deficiencies can undermine the helpfulness of AI conversational models. Though previous research has tried to mitigate them in general domain conversation (Shuster et al., 2021), research to address the entity-level hallucination in KGC remains in dark (Zhang et al., 2022).

To address these entity-level hallucination problems, we propose a post-hoc utterance refining method by entity mining, called **REM**, for more desirable and source-faithful KGC. REM can be used to refine the unfaithful utterances generated by previous models in a plug-and-play manner. To refine utterances, we leverage the entity mining method, which extracts the named entities to implicitly utilize key information in the knowledge in a multi-tasking manner. With this simple but effective method, REM aims to mitigate entity-level hallucination and lead to a more successful conversation.

In order to measure the effectiveness of REM, we conduct an extensive empirical evaluation. First, to demonstrate the post-hoc refining ability of REM, we experiment with refining the utterances generated by baseline models on three KGC datasets. We investigate the flexibility and validity of REM with cross-data experiments and adversarial data refining experiments, respectively. In addition, we conduct an ablation study and human evaluation to verify the effectiveness of our method, showing the improvement of the source-faithfulness score and entity coverage of refined utterances. We also demonstrate the scalability of REM by applying it to large language models. From a case study comparing the refined results of REM and baseline utterance, we demonstrate that our REM model improves the source-faithfulness in the utterances.

Our contributions are threefold: (1) We propose a post-hoc refining method by implicitly mining key entities in the knowledge for more source-faithful conversation; (2) We show that our simple but effective method is adaptable to the existing models, including large language models, in a plug-and-play manner; (3) We substantiate that our method reduces entity-level hallucination and accomplish more desirable knowledge grounded conversation with diverse experiments.

## 2 Related Work

### 2.1 Knowledge Grounded Conversation

In the task of knowledge grounded conversation (KGC), the systems aim to generate an informative conversation based on specialized knowledge. To support research in this area, publicly available datasets for the KGC task have been developed (Gopalakrishnan et al., 2019; Moon et al., 2019; Shuster et al., 2022b). These datasets focus particularly on generating informative conversations on specific topics (Dinan et al., 2018; Zhou et al., 2018; Jang et al., 2022). Building upon these KGC datasets, there has been active research to generate contextually consistent utterances while utilizing the source knowledge. Kim et al. (2020); Adolphs et al. (2021); Wang et al. (2021); Zhang et al. (2023); Feng et al. (2023) incorporate knowledge and context selectors to filter out irrelevant knowledge sentences and redundant dialogue history, respectively. Additionally, (Niu et al., 2023) propose the history-adapted knowledge copy (HAKC) network, which selectively chooses context-aware knowledge to maintain dialogue coherence. Likewise, diverse research is widely conducted in KGC tasks.

### 2.2 Knowledge Hallucination in KGC

Despite the remarkable advancements, KGC systems are known to suffer from knowledge hallucination generating unfaithful utterances to source knowledge (Dziri et al., 2021b). To address hallucination, Shuster et al. (2021) propose neural-retrieval-in-the-loop architectures to improve knowledge-ability consisting of retrievers, rankers, and encoder-decoders. Additionally, Li et al. (2022b) proposes a method that utilizes entity and relation information from a knowledge graph to generate more faithful utterances. However, in KGC, there is a lack of studies on entity-level hallucination, which directly identifies and controls the generation of unfaithful utterances (Das et al., 2023). Although Dziri et al. (2021a) explore the study on entity-level hallucination, this study only considers cases where the source of knowledge is

a knowledge graph. Therefore, there is a need for research on entity-level hallucination in KGC tasks where the source knowledge is provided.

## 2.3 Utterance Refining Methods

Refining methods have been studied to improve the generation results when they are not satisfactory or not in the desired intention. (He, 2021; Geng et al., 2021; Sun et al., 2022; Bao and Zhang, 2023) Especially in dialogue systems, the researches aim to improve the quality of response by applying the refining method have been proposed (Tran and Nguyen, 2018; Posokhov et al., 2022; Wang et al., 2022). To tackle the limitation of generating uninformative or not engaging utterances (Shen et al., 2018), Weston et al. (2018) retrieves the utterance and refines it by regenerating the retrieved utterance. Moreover, for fascinating dialogue systems, the ability to generate personalized responses according to the user's dialogue history is required (Cao et al., 2022). However, as history is usually long and noisy, the problem of missing personalized information exists (Kasahara et al., 2022). To address this problem, Zhong et al. (2022) refines the user dialogue history to extract valuable information. Furthermore, they generate personalized dialogue by utilizing refined history information. Also, Song et al. (2020) adopts a rewriting method for persona-consistent dialogue generation. In this study, our goal is to reduce entity-level hallucination, which is critical in informative conversation (Corbelle et al., 2022), with the refining method.

## 3 Proposed Method

Our proposed REM aims to refine the model-generated utterance to be more faithful to the source knowledge, as shown in Figure 3. The utterance is filtered by the faithfulness scoring function, whether to be refined or not. We train the pre-trained language model with an encoder-decoder structure in the REM method. In the REM model, the entity miner extracts the named entities from the source knowledge and learns them implicitly. The utterance generator makes more faithful utterances based on the key information extracted by the entity miner. In this section, we formulate a KGC task and our proposed utterance refining method, and describe the REM model and its training process.

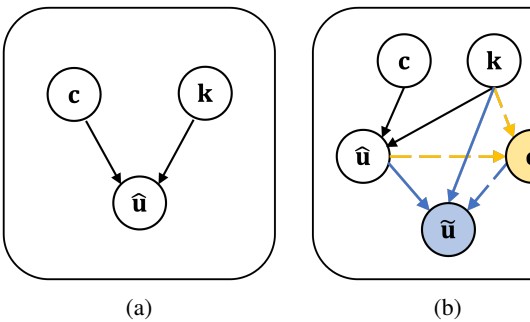

(a)        (b)

Figure 2: The graphical model of (a) the knowledge grounded conversation (KGC) and (b) our proposed REM. The black line is modeled by the KGC model parameter $\theta$. The yellow lines represent the entity miner, which is modeled implicitly, and the blue lines indicates the utterance generator, respectively. The colored lines are modeled by the REM parameter $\psi$.

### 3.1 Knowledge Grounded Conversation

As depicted in Figure 2 (a), the KGC models generate the informative utterance $\hat{u}$ considering both the conversational context (or dialogue history) $c$ and the corresponding source knowledge $k$ as follows:

$$p_\theta(\hat{u}|k, c) = \prod_{t=1}^{n} p_\theta(\hat{u}_t|\hat{u}_{<t}, k, c), \qquad (1)$$

where $n$ indicates the max sequence length of the utterance, and $\theta$ denotes the KGC model parameter.

### 3.2 Post-hoc Utterance Refining

While KGC models aim to generate informative utterances, model-generated utterance $\hat{u}$ may not reflect the input source knowledge faithfully, as shown in Figure 1. In this context, we first filter out the utterance which is not faithful to the source. Then we renovate the given utterance into the refined utterance $\tilde{u}$ which represents the intended knowledge better by utilizing the model-generated utterance $\hat{u}$ and its corresponding knowledge $k$ from KGC models as follows:

$$p_\psi(\tilde{u}|k, \hat{u}) = \prod_{t=1}^{n} p_\psi(\tilde{u}_t|\tilde{u}_{<t}, k, \hat{u}), \qquad (2)$$

where $\psi$ denotes the REM model parameter.

**Faithfulness Filtering** To filter the model-generated utterance $\hat{u}$, we quantify the source-faithfulness score, which indicates how $\hat{u}$ is consistent with the source knowledge $k$. To this end, we adopt the DAE (Goyal and Durrett, 2020) as

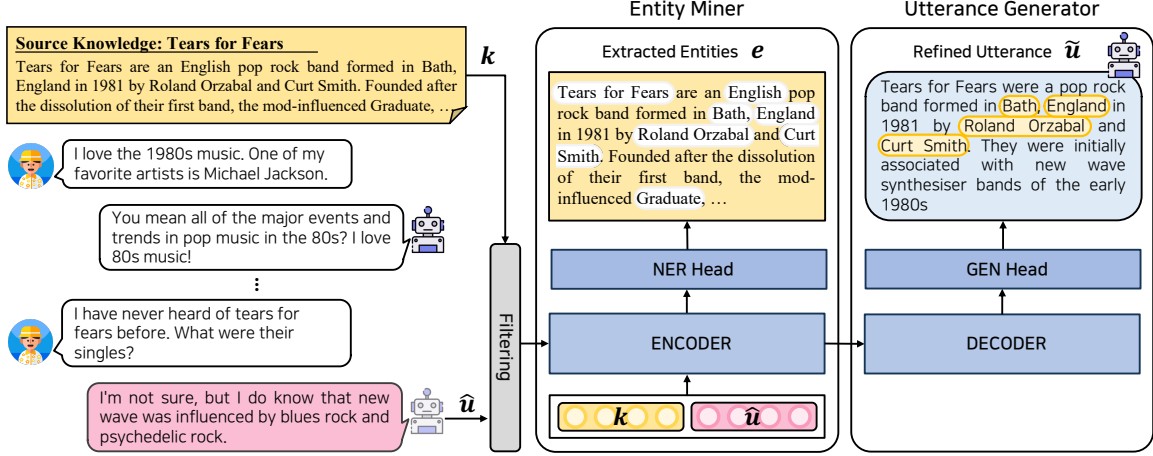

Figure 3: The architecture of REM model, which consists of entity miner and utterance generator. The utterance $\hat{u}$ with a lower source-faithfulness score than threshold $\tau$ enters as input to REM with source knowledge $k$. The entity miner extracts entities $e$ from $k$, in white circles, while the utterance generator refines the $\hat{u}$ with the extracted entity $e$.

a scoring function to estimate the entailment between the source knowledge and $\hat{u}$ considering the dependency arc-level consistency of the utterance.

For efficiency, we only refine the utterances that have lower scores than the threshold according to the source-faithfulness score, as follows:

$$\tilde{u} = \begin{cases} \text{REM}(k, \hat{u}) & : \text{if } score(k, \hat{u}) < \tau \\ \hat{u} & : \text{otherwise,} \end{cases} \quad (3)$$

where $score(\cdot)$ denotes the scoring function (*i.e.*, DAE) and $\tau$ is a threshold value between 0 and 1. When $\tau$ is 1, all utterances are re-generated as $\tilde{u}$; otherwise, when $\tau$ is 0, all utterances are not refined. We utilize this filtering module only in the inference step, not in the training step.

**Refining Utterance by Entity Mining (REM)** Our refining method, REM, is decomposed into two sub-modules that are (i) entity miner and (ii) utterance generator, as shown in Figure 2. The refining task is formulated as follows:

$$p_\psi(\tilde{u}|k, \hat{u}) \propto \underbrace{p_\psi(e|k, \hat{u})}_{\text{Entity miner}} \underbrace{p_\psi(\tilde{u}|e, k, \hat{u})}_{\text{Utterance generator}} \quad (4)$$

These two modules are trained in a multi-tasking manner with parameter $\psi$. The former mines entities $e$ from source knowledge and learns entity-level knowledge implicitly. The latter learns to generate the utterance $u$ with the implicitly mined entities, which is key information of source knowledge,

reducing entity-level hallucinations and producing more source-faithful utterances.

### 3.3 Training Objectives

The training objective of REM mainly falls into named entity recognition (NER) by entity miner and utterance re-generation (GEN) by utterance generator.

To predict the named entities, the NER head is attached on top of the encoder and mines the entities inside the knowledge $k$. It makes the model learn the essential entity information, enhancing the interpretation of the source knowledge. To train NER, we tag named entities with one of the following entity types {‘LOC’,‘PER’,‘ORG’, ‘MISC’} using the off-the-shelf entity tagging module[2] for the source knowledge in the data.

The NER loss $\mathcal{L}_{NER}$ is only defined for the NER label $l_i$ tagged for each token $k_i$ in $k$ and minimized during training as follows:

$$\mathcal{L}_{NER} = -\frac{1}{N_k} \sum_{i=1}^{N_k} l_i \log p(k_i|\hat{u}), \quad (5)$$

where $N_k$ denotes the token length of the source knowledge.

The language modeling head (GEN) learns to refine the given model-generated utterance $\hat{u}$ along with $k$ in an auto-regressive manner while extracting entities $e$. The model learns to compose the information of the inputs and extracted entities from

---

[2] https://github.com/flairNLP/flair

the encoder hidden states. It is trained by minimizing the following loss function:

$$\mathcal{L}_{GEN} = -\frac{1}{N_u} \sum_{i=1}^{N_u} \log p(u_i|u_{<i}, \boldsymbol{k}, \hat{\boldsymbol{u}}; \boldsymbol{e}), \quad (6)$$

where $\boldsymbol{e}$ is implicitly mined in the encoder during training.

The final training objective for a REM trained on both tasks is the following, where $\lambda_n$ is the training hyperparameter:

$$\mathcal{L}_{REM} = \lambda_1 \mathcal{L}_{NER} + \lambda_2 \mathcal{L}_{GEN} \quad (7)$$

## 4 Experimental Settings

In this section, we introduce the dataset, the automatic evaluation metric, and the baseline models used in the experiments. The implementation details are attached in Appendix C.

### 4.1 Datasets

We utilize three datasets as our testbed, but we only use instances for training and testing where ground truth knowledge is given in the data.

FoCus (Jang et al., 2022), which considers both knowledge and persona in conversation, has been released. In FoCus, *machine* generates utterances with customized and knowledgeable utterances about world landmarks. Wizard of Wikipedia (WoW) (Dinan et al., 2018), which is a widely used benchmark, consists of *Wizard* and *Apprentice* talking to each other on various topics in Wikipedia. Another dataset, CMUDoG (Zhou et al., 2018) includes conversations between two speakers discussing different aspects of a specific movie, such as information, plot, etc. The data statistics are presented in Appendix A.

### 4.2 Automated Evaluation Metrics

We evaluate the refining performance of REM with automated metrics in three criteria: source-faithfulness, reference matching, and diversity. To evaluate source-faithfulness, we adopt DAE, entity coverage (EC), and entity type coverage (TC). DAE (Goyal and Durrett, 2020) is used to assess whether the system accurately reflects the facts from the given knowledge when generating utterances at the level of dependency arcs. Furthermore, we utilize two entity-level metrics (EC and TC), following subsequent paragraphs, to evaluate the source-faithfulness in terms of the entity. For the reference matching n-gram score, we utilize

chrF (Popović, 2015), ROUGE-L (Lin, 2004) and SacreBLEU (Post, 2018) to evaluate how close the generated hypothesis is to the ground-truth answer in the test set. For diversity evaluation, Distinct-N[3] (Li et al., 2015) is adopted.

**Entity Coverage and Entity Type Coverage** We compute the entity coverage (EC) and entity type coverage (TC) to quantify the extent of correct entities and context coherence, respectively. To compute EC and TC, we extract the named entities from the generated (model-predicted) utterance $\boldsymbol{u_p}$ (including $\tilde{\boldsymbol{u}}$ and $\hat{\boldsymbol{u}}$), source knowledge $\boldsymbol{k}$, and ground truth utterance $\boldsymbol{u}$, respectively, using the off-the-shelf NER model (Schweter and Akbik, 2020) (The distribution of named entity types are described in Appendix B). First, we evaluate the ratio of the named entities residing in the generated utterance $\boldsymbol{u_p}$ by comparing it with the ground truth utterance $\boldsymbol{u}$ as follows:

$$EC = \frac{\mathcal{N}\big((E_{\boldsymbol{k}} \cap E_{\boldsymbol{u}}) \cap E_{\boldsymbol{u_p}}\big)}{\mathcal{N}\big(E_{\boldsymbol{k}} \cap E_{\boldsymbol{u}}\big)}, \quad (8)$$

where $\mathcal{N}(l)$ is the number of values in list $l$ and $E_{\boldsymbol{k}}$ is the set of named entities in source knowledge. $E_{\boldsymbol{u}}$ and $E_{\boldsymbol{u_p}}$ indicate the set of named entities in ground truth utterance and generated utterance, respectively.

In addition, we evaluate TC, as existing models produce fluent text but demonstrate low context coherence. To identify the intent of context and the information that should be included in the response, we leverage the named entity *types* in the ground truth utterance $\boldsymbol{u}$. For example, if the entity type 'LOC' has the highest proportion among the named entities in $\boldsymbol{u}$, it indicates that the conversation is focused on location-related information. In such cases, the model should generate responses that are relevant to the location to ensure coherence and relevance in the conversation.

To this end, we compute the ratio of name entity types extracted from the generated utterance $\boldsymbol{u_p}$ compared to the ground truth utterance $\boldsymbol{u}$ in each named entity type.

$$TC = \frac{1}{|T|} \sum_t^T \Big(\frac{\mathcal{N}\big(E_{\boldsymbol{u_p}}^t\big)}{\mathcal{N}\big(E_{\boldsymbol{u}}^t\big)}\Big), \quad (9)$$

where $T$ is the set of named entity types {'LOC', 'PER', 'ORG', 'MISC'}.

---

[3] https://github.com/neural-dialogue-metrics/Distinct-N

### 4.3 KGC Baselines

**BART** BART (Lewis et al., 2020a) is Transformer-based (Vaswani et al., 2017) pre-trained model with the encoder-decoder structure. It utilizes various noising methods and in-filling schemes during pre-training. We fine-tune BART on three KGC datasets and use its predicted utterance for train, validation, and test set.

**INFO** INFO (Lim et al., 2022) leverages RAG (Lewis et al., 2020b) to enhance the factuality of the generation results in FoCus dataset. It retrieves a large number of knowledge documents and generates informative conversations based on them.

**EDMem** We utilize and refine the utterances generated by EDMem (Zhang et al., 2022). EDMem is pre-trained on Wikipedia documents to learn entity embeddings and incorporates entity knowledge for entity-intensive dialogue generation.

**ITDD** ITDD (Li et al., 2019) is the best-performing method on the CMUDoG dataset. With a two-pass decoder inspired by the human cognitive process, it improves context coherence and knowledge correctness.

## 5 Results and Analysis

In this section, we report the experimental results and analysis. We first discuss the post-hoc refining ability of our method in §5.1. Then, we evaluate the flexibility of REM through cross-data refining experiments in §5.2. We also conduct experiments on refining adversarial utterances in §5.3, and perform an ablation study in §5.4. Furthermore, we provide the results of human evaluation in §5.5, and present a case study in §5.7. Finally, we analyze the application of REM to large language models in §5.6.

### 5.1 Post-hoc Refining Ability

To demonstrate the post-hoc refining ability, we evaluate the refining performance of REM with vanilla PLM fine-tuned on three datasets and existing models from previous studies. The results are in Table 1 where $REM_{base}$ and $REM_{large}$ refer to the models trained using the REM method on the BART-base and BART-large (Lewis et al., 2020a).

**Fine-tuned Baselines** The comparison results of REM with the vanilla PLM fine-tuned on three

datasets are presented at the top of Table 1. We fine-tune the BART-base model with each dataset and refer to it as $BART_{base}$. To the generated outputs of $BART_{base}$, we evaluate the refining performance with $REM_{base}$. The results demonstrate that the REM leads to improvements not only in the scores of source-faithfulness metrics but also in the reference matching metrics. We also show the ability of REM that refines the utterances of the larger model in Appendix F.

In contrast, REM shows a tendency to decrease the performance of diversity metrics. This indicates that in $BART_{base}$-generated utterances, there are tokens that contribute to increased diversity but also lead to hallucination. These hallucinated tokens are subsequently refined through the REM process. In particular, the significant improvements in EC and TC metrics suggest that entities that cause entity-level hallucination are refined during the refining process with REM.

**Existing Baselines** To investigate the adaptability of REM, we compare the refining performance with previous studies, and the results are annotated with † symbol in Table 1. We re-implement ED-Mem, ITDD, and INFO for fair comparison with $REM_{large}$. The results reveal that $REM_{large}$ exhibits an improvement in source-faithfulness performance. Additionally, it demonstrates enhanced generation performance in the reference matching score, excluding INFO. The reason is that INFO utilizes a large number of knowledge documents for generation, whereas REM generates utterances based on the given ground truth knowledge within the dataset. Furthermore, while the source-faithfulness score increases, the diversity score decreases. This result discusses that REM removes tokens that enhance diversity but also contribute to entity hallucination, similar to the results observed with vanilla PLM.

### 5.2 Cross-data Utterance Refining

To inspect the flexibility of REM, we conduct cross-data experiments, and the results are shown in Table 2.

REM trained on CMUDoG consistently underperforms in source-faithfulness and reference matching scores on the other two datasets. When refining FoCus test set with REM trained on CMU-DoG, the performance significantly decreases in all metrics. This tendency is similarly shown in the results of WoW test set. Likewise, the REM trained

| Data | Model | Source-faithfulness | | | Reference Matching | | | Diversity | |
|---|---|---|---|---|---|---|---|---|---|
| | | EC (%) | TC (%) | DAE | chrF | ROUGE-L | BLEU | Dist-1 | Dist-2 |
| FoCus | BART$_{base}$ | 7.54 | 13.85 | 0.60 | 19.47 | 28.04 | 2.51 | **0.37** | **0.80** |
| | + REM$_{base}$ (Ours) | **23.40** | **20.32** | **0.83** | **30.38** | **33.14** | **9.47** | 0.28 | 0.73 |
| WoW | BART$_{base}$ | 14.23 | 8.80 | 0.84 | 31.70 | 34.02 | 12.72 | **0.31** | **0.78** |
| | + REM$_{base}$ (Ours) | **14.57** | **9.03** | **0.87** | **32.44** | **34.17** | **12.83** | 0.29 | 0.77 |
| CMUDoG | BART$_{base}$ | 3.40 | 3.90 | 0.26 | 12.92 | **13.14** | **3.25** | **0.40** | **0.83** |
| | + REM$_{base}$ (Ours) | **4.95** | **7.55** | **0.40** | **15.06** | 11.92 | 2.49 | 0.25 | 0.70 |
| FoCus$^\dagger$ | INFO | 0.05 | 24.40 | 0.79 | **50.35** | **54.27** | **32.55** | **0.26** | **0.70** |
| | + REM$_{large}$ (Ours) | **0.11** | **25.48** | **0.86** | 49.98 | 52.30 | 30.17 | 0.24 | 0.68 |
| WoW$^\dagger$ | EDMem | 0.00 | 3.50 | 0.50 | 14.30 | 15.21 | 2.90 | **0.28** | **0.71** |
| | + REM$_{large}$ (Ours) | **4.51** | **6.51** | **0.77** | **19.46** | **18.30** | **4.25** | 0.25 | 0.68 |
| CMUDoG$^\dagger$ | ITDD | 0.00 | 1.42 | 0.12 | 9.30 | 9.39 | 1.71 | **0.48** | **0.82** |
| | + REM$_{large}$ (Ours) | **1.01** | **3.96** | **0.30** | **13.64** | **10.44** | **1.74** | 0.24 | 0.67 |

Table 1: Experimental results of baseline utterance refining with REM. The top half shows the results of refining (1) the fine-tuned baselines, and the bottom half shows the results of refining (2) the existing baselines. Numbers in **boldface** indicate higher scores. $^\dagger$ denotes the evaluation settings of the existing models, and the vanilla setting (without $^\dagger$) denotes the evaluation settings of our fine-tuned baseline. The utterances are filtered with $\tau = 0.5$.

| Train | Test | Source-faithfulness | | | Ref. Matching | |
|---|---|---|---|---|---|---|
| | | EC (%) | TC (%) | DAE | chrF | R-L |
| FoCus | | **50.64** | **29.88** | **0.84** | **50.47** | **46.55** |
| WoW | FoCus | 0.03 | 8.35 | 0.03 | 9.24 | 8.65 |
| CMUDoG | | 17.02 | 16.20 | 0.29 | 21.32 | 15.72 |
| WoW | | 20.84 | 13.59 | 0.77 | **37.18** | **31.55** |
| FoCus | WoW | **21.21** | **14.51** | **0.78** | 36.76 | 30.12 |
| CMUDoG | | 18.99 | 12.73 | 0.74 | 35.23 | 29.16 |
| CMUDoG | | 6.90 | 9.50 | 0.43 | **16.42** | 12.44 |
| FoCus | CMUDoG | **12.51** | **29.17** | **0.83** | 13.37 | 7.08 |
| WoW | | 3.40 | 3.90 | 0.26 | 12.92 | **13.14** |

Table 2: The result of cross-data experiments with REM$_{large}$. All utterances are refined with $\tau = 1.0$.

| Data | Model | Source-faithfulness | | | Ref. Matching | |
|---|---|---|---|---|---|---|
| | | EC (%) | TC (%) | DAE | chrF | R-L |
| FoCus | ADV | 27.77 | 19.28 | 0.63 | 27.09 | 31.10 |
| | + REM$_{base}$ | 46.68 | **35.41** | 0.85 | **42.49** | **37.00** |
| | + REM$_{large}$ | **48.04** | 34.19 | **0.87** | 41.91 | 36.87 |
| WoW | ADV | 5.94 | 5.82 | 0.32 | 16.53 | 16.78 |
| | + REM$_{base}$ | **18.93** | **12.71** | 0.75 | 33.03 | **29.33** |
| | + REM$_{large}$ | 17.58 | 12.38 | 0.75 | **34.09** | 28.74 |
| CMUDoG | ADV | 3.33 | 4.98 | 0.23 | 12.07 | **12.04** |
| | + REM$_{base}$ | 5.21 | **10.08** | **0.44** | 15.37 | 10.92 |
| | + REM$_{large}$ | **5.99** | 9.24 | 0.42 | **15.55** | 11.43 |

Table 3: Experimental results of adversarial data refining. 'ADV' denotes the adversarially edited utterances by ChatGPT. All utterances are refined with $\tau = 1.0$.

on WoW exhibits a decrease in performance on FoCus and CMUDoG test sets.

According to the previous comprehensive human study (Dziri et al., 2022b) that analyzes the portion of hallucination in KGC dataset, the reason for the performance decrease is hallucinated utterances in WoW and CMUDoG datasets. The results revealed that only 24.15% of utterances in wow and 16.2% in CMUDoG exhibited entailment. These proportions indicate that a significant portion of utterances contains knowledge hallucination. Consequently, hallucination in the dataset has an impact on model training and evaluation.

On the other hand, the model trained on FoCus dataset demonstrates high source-faithfulness performance across all datasets. We assume that the reason for performance differences lies in dataset construction. While WoW and CMUDoG contain

utterances that provide responses without knowledge, FoCus ensures that all utterances are generated based on knowledge in the dataset. Therefore, the REM trained on FoCus has facilitated the training of a more faithful KGC model.

## 5.3 Adversarial Data Refining

To investigate the validity of REM, we have prompted ChatGPT (OpenAI-Blog, 2022) to generate the synthetic data by adversarially changing the ground truth utterance[4]. We demonstrate the performance of REM by refining the utterances that are distorted to be unfaithful to the source knowledge; all data are refined with ($\tau = 1.0$). As presented in Table 3, compared to the adversarially synthesized

---

[4]The prompts are shown in Appendix E.

| Model | Source-faithfulness | | | Ref. Matching | |
|---|---|---|---|---|---|
| | EC (%) | TC (%) | DAE | chrF | R-L |
| BART$_{base}$ | 19.47 | 28.04 | 7.54 | 13.85 | 0.60 |
| + REM$_{base}$ wo NER | 30.24 | 33.08 | 23.28 | 20.22 | 0.82 |
| + REM$_{base}$ | **30.38** | **33.14** | **23.40** | **20.32** | **0.83** |
| + REM$_{large}$ wo NER | 31.30 | 34.44 | **24.17** | 20.26 | 0.82 |
| + REM$_{large}$ | **31.52** | **34.56** | 20.58 | **20.58** | **0.84** |

Table 4: The result of ablation study on training objectives. The utterances are filtered with $\tau = 0.5$

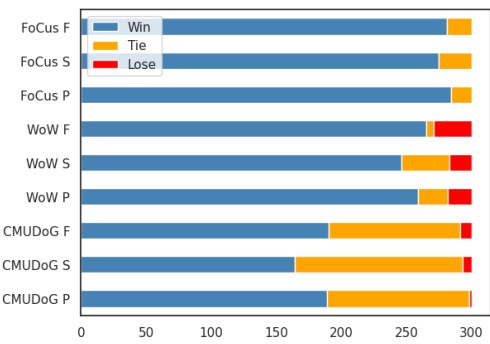

Figure 4: Results of human evaluation. F, S and P, attached after the dataset name, denote fluency, source faithfulness and paraphrasing, respectively.

data (ADV), both source-faithfulness and reference matching scores increased significantly after refining. ROUGE-L score of the ADV is slightly higher in CMUDoG, which is likely due to the characteristic that the utterances of CMUDoG have much shorter lengths than other datasets (Dziri et al., 2022a).

## 5.4 Ablation Study

To explore the effect of entity miner, we conduct an ablation study of our proposed method. In Table 4, we present the results on the FoCus dataset, which has the most informative conversations in §5.2. We compare the REM model trained with multi-task learning and only trained with utterance refining (w/o NER).

In case the model is trained without entity miner, the results show a performance decrease across the board except in one case. Our proposed REM method, which learns essential information with entity miner, can be demonstrated to be effective for source-faithfulness and reference matching scores. The DAE score of the *large* models shows better performance without entity miner. The large model shows unstable results in §5.3. However, a significant improvement in the performance of the REM model is shown, with or without entity miner, compared to the baseline utterance. This suggests that refining helps increase the quality and source-faithfulness of fatal utterances.

## 5.5 Human Evaluation

To qualitatively evaluate the results before and after refining, we perform a human evaluation on 100 randomly sampled utterance examples, each from three datasets. One example contains an utterance produced by the baseline and the refined utterance by REM, and we evaluate them with three criteria: 1) fluency, 2) source-faithfulness, and 3) paraphrasing. The first criterion assesses naturalness by measuring the fluency of the generated results. The

second criterion measures source-faithfulness by assessing whether the sentence is factually consistent with the given knowledge. In addition, the third criterion evaluates whether the given knowledge has been appropriately reorganized and incorporated into the utterance rather than just copied and pasted. We provide the questionnaire used in Appendix G

The results of the comparative human evaluation of baseline utterances with scores below the threshold and utterances after refining are shown in Figure 4 ($\tau = 0.5$). Across all data and criteria, the refined utterances win in most cases. Especially in FoCus, refined utterances do not lose to the utterance before refining. REM performs the worst in CMUDoG, which is similar to the experiment in Section 5.1. The dissimilarity between the CMUDoG and the other two datasets can be attributed to its predominant resemblance to a general utterance instead of the knowledge grounded conversations. Nevertheless, we qualitatively demonstrate the effectiveness of REM with a significantly higher number of winning cases.

## 5.6 REM-LLM

To show the scalability of our method to large language models (LLMs), we apply REM method to the *prompt* of not-tunable LLM (*i.e.*, ChatGPT) beyond fine-tuning the models. In this experiment, large language models with (LLM$_{REM}$) and without (LLM) the REM method refine the utterance of the KGC baseline model (BART$_{base}$) and Chat-GPT (LLM$_{KGC}$). LLM is required to refine the utterance considering the given source knowledge, while LLM$_{REM}$ is made to modify the utterance by extracting key entities from the knowledge and utilizing them. We attach the prompts used in Ap-

| Data | Model | Source-faithfulness | | | Ref. Matching | |
|------|-------|-----|-----|-----|------|-----|
| | | EC | TC | DAE | chrF | R-L |
| FoCus | $\text{BART}_{base}$ | 7.56 | 13.85 | 0.60 | 19.47 | 28.03 |
| | + LLM | 23.54 | 20.14 | 0.86 | 26.96 | 33.57 |
| | + $\text{LLM}_{REM}$ | **25.28** | **22.18** | **0.89** | **28.27** | **33.66** |
| WoW | $\text{BART}_{base}$ | 14.23 | 8.79 | 0.84 | 31.73 | 34.02 |
| | + LLM | 14.34 | 8.89 | 0.85 | 31.95 | **33.87** |
| | + $\text{LLM}_{REM}$ | **14.57** | **9.12** | **0.86** | **32.27** | 33.84 |
| CMUDoG | $\text{BART}_{base}$ | 3.40 | 3.90 | 0.26 | 12.92 | **13.14** |
| | + LLM | 4.93 | 6.07 | 0.36 | 13.26 | 12.43 |
| | + $\text{LLM}_{REM}$ | **7.36** | **11.39** | **0.57** | **14.70** | 11.23 |
| FoCus | $\text{LLM}_{KGC}$ | 42.74 | 26.72 | 0.84 | 38.16 | 40.64 |
| | + LLM | 44.95 | 27.45 | 0.89 | 39.61 | 39.61 |
| | + $\text{LLM}_{REM}$ | **45.83** | **28.46** | **0.91** | **40.01** | **41.98** |
| WoW | $\text{LLM}_{KGC}$ | 6.91 | 5.87 | 0.36 | 18.68 | 18.65 |
| | + LLM | 9.27 | 6.73 | 0.47 | 21.24 | 20.17 |
| | + $\text{LLM}_{REM}$ | **13.28** | **9.25** | **0.63** | **25.17** | 21.19 |
| CMUDoG | $\text{LLM}_{KGC}$ | 4.77 | 7.24 | 0.29 | **14.36** | **13.44** |
| | + LLM | 5.61 | 8.53 | 0.38 | **14.36** | 12.21 |
| | + $\text{LLM}_{REM}$ | **7.88** | **14.26** | **0.54** | **14.36** | 10.04 |

Table 5: The result of REM with the large language model (denoted with +) refining the KGC utterances of $\text{BART}_{base}$ and large language model ($\text{LLM}_{KGC}$). The utterances are filtered with $\tau = 0.5$

pendix H.

As shown in Table 5, both LLM and $\text{LLM}_{REM}$ show improved source-faithfulness performance, especially in CMUDoG and FoCus. $\text{LLM}_{REM}$ is able to achieve the most significant improvement by entity mining and refining the utterance with key entities. This demonstrates that when prompting with LLMs, more source-faithful utterances can be produced with REM by asking LLMs to find the important parts and use them for making responses.

### 5.7 Case Study

To analyze the generative results of REM, we present the examples of three datasets in Table 6. In FoCus example, the utterance of baseline answers only provides the information of *year*, but with the imperfect entity mention 'Per'. However, after being refined with REM, the utterance provides more specific knowledge with correct entity mentions, even utilizing the persona knowledge. In WoW example, the baseline utterance provides knowledge that is not given in the knowledge, but REM gives source-faithful information. In CMUDoG example, REM provides more detailed factual knowledge in the utterance than in the baseline utterance. We show more examples also in Appendix I.

---

**FoCus**

**[Knowledge]**
Designed by French architect Paul Andreu, the NCPA opened in 2007 and is the largest theatre complex in Asia.
**[Persona Knowledge]**
I love art. I'm interested in architecture. I love lakes. I want to go on a trip. I want to visit China.
**[Utterance]**
$\text{BART}_{base}$ : National for Per Arts in opened 2007.
**+ $REM_{base}$ : National Center for the Performing Arts (NCPA) opened in 2007 and is the largest theatre complex in Asia. If you love art, you have to come see it with your own eyes!**

---

**WoW**

**[Knowledge]**
Among other honors, he has won nine Goyas, two European Film Awards and an Oscar.
**[Utterance]**
$\text{BART}_{base}$ : I've seen it a few times. It was directed by Alejandro Amená
**+ $REM_{base}$ : of course! He has won nine Goyas, two European Film Awards, and an Oscar! That's a lot of awards for a director!**

---

**CMUDoG**

**[Knowledge]**
The Asgardian Loki encounters the Other, the leader of an extraterrestrial race known as the Chitauri. ..., Loki takes the Tesseract and uses his scepter to enslave Selvig and a couple of other agents, including Clint Barton, to aid him in his getaway.
**[Utterance]**
$\text{BART}_{base}$ : you should watch the avengers 2 and 3
**+ $REM_{base}$ : you should watch the avengers 2 and 3, they are really good as well. what did you think of the scene where Loki takes the Tesseract?**

---

Table 6: Examples of refined utterances by REM on three datasets.

## 6 Conclusion

In this work, we proposed REM, a post-hoc refining method for improving the source-faithfulness of the utterances in the knowledge grounded conversation (KGC). REM enabled simple but effective refining by extracting entities from the source knowledge given in the KGC task. It makes the model learn the key information implicitly and use it for refining the utterance more faithful to the source knowledge. We presented extensive experiments applying REM in a plug-in-play method to various model-generated outputs, showing increased source-faithfulness and entity coverage after refining. Also, qualitative analysis and human evaluation proved the refining efficacy of REM. We verified that our proposed method could be utilized with the prompt of the large language models.

## Limitations

Our research proposed a method that aims to refine the non-source-faithful utterances in the knowledge grounded conversation. Though we address the problem of models not reflecting even the ground truth knowledge given, it is difficult to solve if the retriever does not give accurate knowledge from the in-the-wild setting. Therefore, the performance of the refiner may depend on the retriever's performance. As it is beyond the scope of this paper, we leave it as future work. When filtering the utterances by source-faithfulness score, we only utilized DAE, but other scores or classifiers (Dou et al., 2022; Manakul et al., 2023) can be adopted for filtering according to the data or domain. There is an area of research in detecting *critical errors* in generated results in neural machine translation. Similarly, in the dialogue tasks, the better the classifier performs in detecting hallucinated utterances, the better the performance of the refiner will be. In addition, we expect that existing baseline models could be further improved if trained with REM in an end-to-end manner, so we leave it as future work.

## Ethics Statement

The datasets used in our work are from previously published papers, so we do not attach privacy or ethical issues to the dataset. At inference, we will make the model not generate tokens in a list of several stopwords utilizing Hatebase[5], Silva et al. (2016), etc., to avoid generating harmful utterances that the model may have learned during pre-training. As we are aware that excessive computational energy used to train models stimulates environmental problems, we adopt the multi-task learning method for training entity miner and utterance generator instead of having a separate entity miner model to improve efficiency. Also, we distribute all codes and model checkpoints so they do not have to be trained again. We believe that our refining method can contribute to mitigating the entity-level hallucination of model-generated utterances and preventing the misuse of AI systems.

## Acknowledgements

This research was supported by the MSIT(Ministry of Science and ICT), Korea, under the ITRC(Information Technology Research Center)

---

[5]http://hatebase.org

support program(IITP-2023-2018-0-01405) supervised by the IITP(Institute for Information & Communications Technology Planning & Evaluation). This work was supported by Institute of Information & communications Technology Planning & Evaluation(IITP) grant funded by the Korea government(MSIT) (No. 2020-0-00368, A Neural-Symbolic Model for Knowledge Acquisition and Inference Techniques). Also, this work was supported by NCSOFT NLP Center.

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

## A    Dataset Details

FoCus (Jang et al., 2022) is a dataset where a *human* and a *machine* take turns having a conversation about a specific landmark. This dataset consists of 14,452 conversations, with 173,424 utterances. We use the validation set, which has 1,445 conversations and 17,340 utterances, for the experiment, as the official test set does not provide ground truth knowledge and utterances.

Wizard of Wikipedia (WoW) (Dinan et al., 2018) is a conversational dataset based on Wikipedia articles on various topics. The dataset covers 1,365 topics and consists of 22,311 conversations with a total of 201,999 utterances, which are divided into 166,787 for train, 17,715 for validation, and 17,497 for test. We used a random split version of the validation and test set. Only for evaluating ED-Mem (Zhang et al., 2022), we follow its test setting, KILT (Petroni et al., 2021), a variant of the WoW dataset.

CMUDoG (Zhou et al., 2018) is a knowledge grounded conversation dataset with two speakers conversing based on movie Wikipedia articles. Unlike the WoW and FoCus, where only one of the two speakers has access to the knowledge, both speakers of this dataset have access to the content of the article. It has a resemblance to a generic dialogue compared to the other two datasets. It is composed of a total of 4,112 conversations with an average of 21.43 turns.

## B    Distribution of Named Entity-Type

We show the entity type distribution of all datasets, the train, development, and test sets used for fine-tuning the baseline model $BART_{base}$, in Figure 5. WoW has the evenest distribution, and FoCus has the second most even distribution. CMUDoG has the most "PER" entity class among the other three classes.

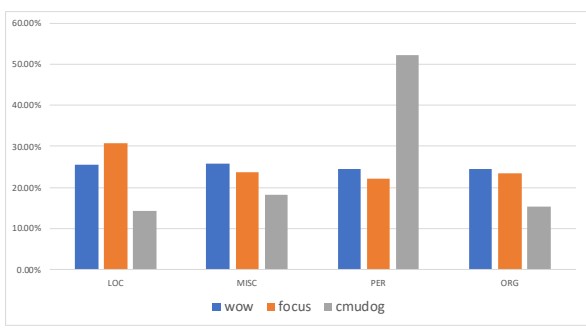

Figure 5: Entity type distribution of three datasets.

## C    Implementation Details

When implementing REM, we adopt BART (Lewis et al., 2020a), which has an encoder-decoder structure, and train the entity mining and utterance generation tasks. We experiment with $BART_{base}$ of 140M parameters and $BART_{large}$ of 406M parameters. We train REM on the pairs of model-predicted utterances, generated by the fine-tuned BART baseline, and reference utterance. We implement the models by exploiting Pytorch (Paszke et al., 2019) and HuggingFace (Wolf et al., 2019) with a fixed seed for reimplementation. The models are trained with a learning rate of 6.25e-5 for 10 epochs with early stopping. AdamW (Loshchilov and Hutter, 2017) is used as the optimizer. We set the train batch size to 8 with a gradient accumulation of 32. The time required for training is about 2 hours per epoch on a single RTX-6000 GPU. For decoding, we used a beam size of 5, min length of 32, max length of 512, top $k$ of 50, and no-repeat n-gram size of 2. NER loss weight $\lambda_1$ and GEN loss weights $\lambda_2$ are set as 0.5:1, 0.3:1, and 0.7:1 for WoW, CMUDoG, and FoCus, respectively, according to the preliminary study. We will open the source codes which we used for the experiments after the review process.

| Data | Model | ACC (NER) | F1 (ACC) | PPL (GEN) |
|------|-------|-----------|----------|-----------|
| FoCus | + $REM_{base}$ | 0.991 | 0.944 | 3.778 |
| FoCus | + $REM_{large}$ | 0.989 | 0.925 | 3.323 |
| WoW | + $REM_{base}$ | 0.998 | 0.938 | 12.742 |
| WoW | + $REM_{large}$ | 0.997 | 0.913 | 10.788 |
| CMUDoG | + $REM_{base}$ | 1.000 | 1.000 | 29.775 |
| CMUDoG | + $REM_{large}$ | 0.999 | 0.991 | 27.247 |

Table 7: The result on the validation set used for training and evaluating the models with both tasks, NER and GEN. Larger models show slightly higher scores in GEN, but lower scores in ACC.

## D    Threshold

To utilize the off-the-shelf metric as our scoring function for filtering, we manually find the threshold both for effectiveness and efficiency. The high threshold is intended for evaluating the source-faithfulness of a given utterance strictly. A low threshold, on the other hand, targets only those utterances that are not faithful to the knowledge and aims to regenerate only fatal cases. As shown in Figure 6, we use a threshold of 0.5 to consider all datasets and 1.0 if we need to refine all utterances.

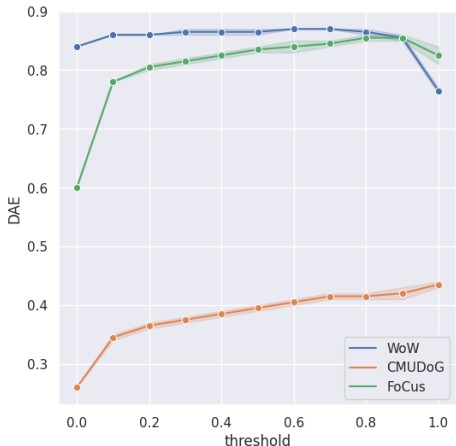

Figure 6: Manual search for the threshold of filtering module. DAE score in the y-axis indicates the scource-faithfulness score after refining the utterances under the threshold.

## E Prompts for Adversarial Data Generation with ChatGPT

The prompt used for generating adversarial data with ChatGPT is in Table 8:

| Prompt |
| --- |
| "Considering the knowledge, persona, and dialogue history below and make the answer incorrect in terms of factuality. Knowledge: {knowledge}, dialogue history: {dialogue_history}, answer: {answer}." |

Table 8: The prompt used for generating adversarial data.

## F Refining Utterances of Larger Models

| Data | Model | Source-faithfulness | | | Ref. Matching | |
| --- | --- | --- | --- | --- | --- | --- |
| | | EC | TC | DAE | chrF | R-L |
| FoCus | LLM$_{KGC}$ | 42.74 | 26.72 | 0.84 | 38.16 | 40.64 |
| | + BART$_{REM_{large}}$ | **46.06** | **28.76** | **0.91** | **40.95** | **42.36** |
| WoW | LLM$_{KGC}$ | 6.91 | 5.87 | 0.36 | 18.68 | 18.65 |
| | + BART$_{REM_{large}}$ | **15.97** | **11.15** | **0.77** | **31.08** | **27.44** |
| CMUDoG | LLM$_{KGC}$ | 4.77 | 7.24 | 0.29 | 14.36 | **13.44** |
| | + BART$_{REM_{large}}$ | **6.27** | **9.63** | **0.43** | **16.10** | 12.38 |

Table 9: The result of BART$_{REM_{large}}$ (denoted with +) refining the KGC utterances of large language model (LLM$_{KGC}$). The utterances are filtered with $\tau = 0.5$

## G Human Evaluation Questionnaire

| Questionnaire |
| --- |
| The given sentences from model A and model B are the generated result of each machine consulting its knowledge to produce informative utterances. Considering the sentences produced by model A and model B, name the model that did better on the following three criteria. **Fluent** : The response is linguistically fluent and not awkward). **Factually correct** : The response is consistent with the given knowledge and does not generate the facts that are not given. **Well-paraphrased** : The response is well-paraphrased rather than just copying and pasting the given knowledge. |

Table 10: An example of the questionnaire for human evaluation.

## H Prompts for LLM and LLM$_{REM}$

The prompts for refining utterances with LLM and LLM$_{REM}$ are in Table 11.

| LLM |
| --- |
| "You are tasked with refining the response given the knowledge, question, and response. First, you determine whether the response is a factual response given the knowledge and question. If the response is a factual response considering the given knowledge and question, output it as it is, and if not, regenerate it after factually refining the response considering the given knowledge and question. You should only write "Output" in the format given without saying why. Knowledge: {knowledge} Response: {response}." |

| LLM$_{REM}$ |
| --- |
| "You are tasked with refining the response given the knowledge, question, and response. You determine whether the response is a factual response considering the given knowledge and question, and if not, extract the necessary entities to refine the response more factually from the knowledge. Then, refine and regenerate the response into a factual response considering the given knowledge and question using the extracted entities. You must write both "Entities" and "Output" and only "Entities" and "Output" according to the given format. Knowledge: {knowledge} Response: {response}" |

Table 11: The prompts for refining utterance by LLM and LLM$_{REM}$.

# I REM Generation Examples

| **FoCus** |
|---|
| **[Knowledge]** |
| The Václav Havel Library, located in Prague, is a charitable organization founded by Dagmar Havlová, Karel Schwarzenberg and Miloslav Petrusek on 26 July 2004. |
| **[Persona Knowledge]** |
| I love going to libraries. I love reading. I want to visit Prague. I want to visit New York. I love books. |
| **[Utterance]** |
| $BART_{base}$ : cl Ha Library founded 26 2004. |
| **+ $REM_{base}$ : Václav Havel Library was founded on 26 July 2004 by Dagmar Havlová, Karel Schwarzenberg and Miloslav Petrusek.** |
| **[Knowledge]** |
| The design of the house is attributed to John Verge and John Bibb and was built from 1835 to 1839 by James Hume. |
| **[Persona Knowledge]** |
| I love colonial architecture. I like Australia. I would like to visit a museum. I would love to see a grotto. I have read about the Cadigal people. |
| **[Utterance]** |
| $BART_{base}$ : was from35 18 to39 |
| **+ $REM_{base}$ : it was built from 1835 to 1839 by James Hume. I got this information on the internet. It was designed by John Verge and John Bibb.** |

| **WoW** |
|---|
| **[Knowledge]** |
| A tattoo is a form of body modification where a design is made by inserting ink, dyes and pigments, either indelible or temporary, into the dermis layer of the skin to change the pigment. |
| **[Utterance]** |
| $BART_{base}$ : I'm not sure if it is really a tattoo artist, but I do know that |
| **+ $REM_{base}$ : of course, it is a form of body modification where a design is made by inserting ink, dyes and pigments into the dermis layer of the skin to change the pigment.** |
| **[Knowledge]** |
| It was also applied to the pit-style cooking techniques now frequently used in the Southeastern United States. |
| **[Utterance]** |
| $BART_{base}$ : , I'll have to try it sometime! I've heard it's used in the |
| **+ $REM_{base}$ : , I don't know if I'd ever try it. I know that pit-style cooking techniques are now frequently used in the Southeastern United States.** |

| **CMUDoG** |
|---|
| **[Knowledge]** |
| Leonardo DiCaprio as Dom Cobb, a professional thief who specializes in conning secrets from his victims by infiltrating their dreams. Joseph Gordon-Levitt as Arthur, Cobb's partner who manages and researches the missions. ... Tom Hardy as Eames, a sharp-tongued associate of Cobb. wildly ingenious chess game, ... |
| **[Utterance]** |
| $BART_{base}$ : , have you seen the movie inception? |
| **+ $REM_{base}$ : , I'm good, how are you? Have you seen Inception? It's a science fiction movie starring Leonardo DiCaprio, Joseph Gordon-Levitt, and Tom Hardy.** |
| **[Knowledge]** |
| Angelina Jolie as Maleficent Sharlto Copley as Stefan Elle Fanning as Aurora Sam Riley as Diaval Imelda Staunton as Knotgrass Maleficent' demolishes stereotypes ... Robert Stromberg SuperHero Maleficent is a 2014 American dark fantasy film directed by Robert Stromberg from a screenplay by Linda Woolverton, and starring Angelina Jolie as Maleficent with Sharlto Copley, Elle Fanning, Sam Riley, Imelda Staunton, Juno Temple, and Lesley Manville in supporting roles. ... |
| **[Utterance]** |
| $BART_{base}$ : have you seen Maleficent before? |
| **+ $REM_{base}$ : movie we're talking about is Maleficent. Have you seen it? It's a fantasy from 2014. It stars Angelina Jolie, Sharito Copley, Elle Fanning, Sam Riley, and Imelda Staunton.** |

Table 12: Examples of the fine-tuned baseline utterances refining by $REM_{base}$ on KGC datasets.

| FoCus |
|---|

**[Knowledge]**
Elizabeth Bay House is a heritage-listed Colonial Regency style house and now a museum and grotto, located at 7 Onslow Avenue in the inner eastern Sydney suburb of Elizabeth Bay in the City of Sydney local government area of New South Wales, Australia.
**[Persona Knowledge]**
I love colonial architecture. I like Australia. I would like to visit a museum. I would love to see a grotto. I have read about the Cadigal people.
**[Utterance]**
$BART_{base}$ : Elizabeth House located Australia which like. is place you like
+ $REM_{large}$ : **This place is called Elizabeth Bay House and it is located in Australia, which you like. It is a heritage-listed Colonial Regency style house and now a museum and grotto.**

**[Knowledge]**
The Memphis Zoo, located in Midtown, Memphis, Tennessee, United States, is home to more than 3,500 animals representing over 500 different species.
**[Persona Knowledge]**
I hope to visit Tennessee. I like to ride on a tram. I have never seen a wetland. I would like to see a Giant Panda. I wish I could see a penguin.
**[Utterance]**
$BART_{base}$ : is Memphis in. is place hope visit.
+ $REM_{large}$ : **It's the Memphis Zoo in Tennessee, a place you hope to visit. It is home to more than 3,500 animals, 500 different species.**

| WoW |
|---|

**[Knowledge]**
Competing theories state that Hindu identity developed in the British colonial era, or that it developed post-8th century CE after the Islamic invasion and medieval Hindu-Muslim wars.
**[Utterance]**
$BART_{base}$ : , I'm not fluent in English, but I understand that Hindu identity developed in the
+ $REM_{large}$ : **No, I'm not fluent in English, but I understand that Hindu identity developed in the British colonial era, or that it developed post-8th century CE.**

**[Knowledge]**
Human occupation of Qatar dates back to 50,000 years ago, and Stone Age encampments and tools have been unearthed in the peninsula.
**[Utterance]**
$BART_{base}$ : 's a pretty busy place, I've been to Qatar a few times. It's
+ $REM_{large}$ : **That's true. It's a pretty busy place, but it's been around for a long time. Human occupation of Qatar dates back 50,000 years ago, and Stone Age encampments and tools have been found in the peninsula.**

| CMUDoG |
|---|

**[Knowledge]**
Sally Hawkins as Elisa Esposito, a mute cleaner who works at a secret government laboratory. ... The Shape of Water Rotten Tomatoes: 92% and average: 8.4/10 Metacritic Score: 87/100 CinemaScore: A 2017
**[Utterance]**
$BART_{base}$ : have you seen The Shape of Water?
+ $REM_{large}$ : **Hello, Have you seen The Shape of Water? It's a really good movie. It has a 92% on Rotten Tomatoes and a Metacritic Score of 87/100.**

**[Knowledge]**
Lindsay Lohan as Cady Heron Rachel McAdams as Regina George Lacey Chabert as Gretchen Wieners Amanda Seyfried as Karen Smith Lizzy Caplan as Janis Ian Boasts a one-two-three punch in star Lindsay Lohan, screenwriter Tina Fey and director Mark Waters, and, indeed, it delivers a knockout The screenplay was highly praised by critics with Peter Travers of Rolling Stone calling it 'comic gold'. ...
**[Utterance]**
$BART_{base}$ : , what did you think of it?
+ $REM_{large}$ : **I really liked it, I thought Lindsay Lohan did a great job as Cady Heron. Tina Fey did an amazing job writing the screenplay as well.**

Table 13: Examples of the fine-tuned baseline utterances refining by $REM_{large}$ on KGC datasets.

| FoCus |
|---|

**[Knowledge]**

hesselberg ( pronounced [ ] ( about this soundlisten ) ; 689 m above sea level ) is the highest point in middle franconia and the franconian jura and is situated 60 km south west of nuremberg, germany.

**[Persona Knowledge]**

I love to enjoy nature. I live near Nuremberg. I am from Germany. I like heights. I study geology.

**[Utterance]**

*INFO* : The name of this place is Hesselberg.

+ *REM*$_{large}$ **: The name of this place is Hesselberg, and it is located 60 km south west of nuremberg, germany, the country you are from.**

---

**[Knowledge]**

st. george's united methodist church, located at the corner of 4th and new streets, in the old city neighborhood of philadelphia, is the oldest methodist church in continuous use in the united states, beginning in 1769.

**[Persona Knowledge]**

I will stay in Philadelphia. I am going to Pennsylevania. I like history. I am religious. I am interested in the American Revolution.

**[Utterance]**

*INFO* : This is St. George's United Methodist Church you may know of it because you are religious.

+ *REM*$_{large}$ **: This is St. George's United Methodist Church, you may know of it because you are religious. It is the oldest methodist church in continuous use in the united states.**

| WoW (KILT) |
|---|

**[Knowledge]**

Red Ketchup is a cult Quebec comic book series featuring FBI's crazed rogue agent, Steve ŘedǨetchup. The series was created by Pierre Fournier and Réal Godbout, and will soon be adapted into a live action feature film by Martin Villeneuve for GO Films in Montreal.

**[Utterance]**

*EDMem* : ketchup red is a sweet sauce made from tomatoes.

+ *REM*$_{large}$ **: I love red ketchup, which is a cult Quebec comic book series featuring FBI's crazed rogue agent, Steve ŘedǨetchup. The series was created by Pierre Fournier and Réal Godbout**

---

**[Knowledge]**

Childhood obesity is a condition where excess body fat negatively affects a child's health or well-being. As methods to determine body fat directly are difficult, the diagnosis of obesity is often based on BMI. Due to the rising prevalence of obesity in children and its many adverse health effects it is being recognized as a serious public health concern. The term overweight rather than obese is often used when discussing childhood obesity, especially in open discussion, as it is less stigmatizing.

**[Utterance]**

*EDMem* : i'm not sure, but i do know that in the united states, overweight people are more likely to be obese than healthy people.

+ *REM*$_{large}$ **: Yea i'm not sure, but the prevalence of obesity in children and its many adverse health effects it is being recognized as a serious public health concern.**

| CMUDoG |
|---|

**[Knowledge]**

Leonardo DiCaprio as Frank Abagnale, Jr. Tom Hanks as Carl Hanratty Christopher Walken as Frank Abagnale, Sr. Nathalie Baye as Paula Abagnale. ... Catch me if you can Rotten Tomatoes: 96% and average: 7.9/10 Metacritic Score: 76/100 CinemaScore: A- 2002

**[Utterance]**

*ITDD* : yes, it has a 96 % rating on rotten tomatoes.

+ *REM*$_{large}$ **: yes, it has a 96 percent rating on rotten tomatoes and 7.9/10 on metacritic and a cinema score of an A- which is pretty good.**

---

**[Knowledge]**

Mike and Sulley are led away by the CDA, and are expelled from the university as a result of their actions, but the other members of Oozma Kappa are accepted into the scare program the next semester, as Hardscrabble was impressed by their performances in the Games. As Mike leaves on the bus, Sulley runs after him to raise his spirits. Hardscrabble then appears and wishes the two luck, claiming they were the first students to have surprised her. The two take jobs in the mail room of Monsters, Inc., eventually working their way up to join the scarer team.

**[Utterance]**

*ITDD* : they do

+ *REM*$_{large}$ **: Yes, they do. Sulley is a good friend of Mike's and tries to cheer him on when he leaves on the bus to try to raise his spirits.**

Table 14: Examples of the existing model utterances refining by REM$_{large}$ on KGC datasets.