# OpenReview forum: "Post-hoc Utterance Refining Method by Entity Mining for Faithful Knowledge Grounded Conversations"
_EMNLP/2023/Conference — EMNLP 2023 Main_

### Official Review · Reviewer_9ska · 2023-07-24

**Soundness:** 3

**Excitement:**

3: Ambivalent: It has merits (e.g., it reports state-of-the-art results, the idea is nice), but there are key weaknesses (e.g., it describes incremental work), and it can significantly benefit from another round of revision. However, I won't object to accepting it if my co-reviewers champion it.

**Paper Topic And Main Contributions:**

This paper proposes REM, which uses an encoder-decoder model with named entity recognition (NER) to refine a language model response for increased faithfulness to provided source knowledge. The encoder is trained to extract named entities from the source knowledge; jointly, the decoder is trained to refine the original response given the source knowledge and extracted entities from the encoder. REM improves the entity coverage of model responses.

**Questions For The Authors:**

A: For the LLM experiment (Section 4.6), are the LLMs just used to refine the BART responses? What happens when the LLMs generate their own responses from scratch, and REM is used to refine LLM responses? E.g. it would be interesting (but may not be the case) if REM (BART-large, 400M parameters) is still able to improve the responses of a larger model.

**Reasons To Accept:**

REM improves the entity coverage of model responses relative to source knowledge. The ablation study demonstrates that the NER sub-task is helpful for REM.

**Reasons To Reject:**

It seems that the approach is only tested on refining BART-large model responses (400M parameters). It is unclear whether the approach would still be effective for refining LLM responses (e.g. open-source GPT-style models with >1-10B parameters, or responses from API calls to larger LLMs).

Discussion period update: the authors have presented results for GPT-4, demonstrating that their REM method can still improve the factuality of responses. This is useful given the widespread usage of these larger models for chat systems (excitement score increased to 3).

I agree with other reviewers that much of the related work and implementation sections in the appendix should appear in the main text.

**Reproducibility:**

4: Could mostly reproduce the results, but there may be some variation because of sample variance or minor variations in their interpretation of the protocol or method.

**Reviewer Confidence:**

3: Pretty sure, but there's a chance I missed something. Although I have a good feel for this area in general, I did not carefully check the paper's details, e.g., the math, experimental design, or novelty.

---

> ### Author Rebuttal · Authors · 2023-08-25
>
> Thanks for your constructive comments. We sincerely appreciate your time in reading the paper. Please find the responses below.
>
> &nbsp;
>
>     Reasons To Reject: It seems that the approach is only tested on refining BART-large model responses (400M parameters). It is unclear whether the approach would still be effective for refining LLM responses (e.g., open-source GPT-style models with >1-10B parameters, or responses from API calls to larger LLMs).
>
> Re: We believe that the reason for the rejection is outside the scope of our paper, as our goal was not to use the smaller model to correct the larger model's responses or to correct all the answers, but to refine only the erroneous utterances in a post-hoc manner to reduce the entity hallucination during the knowledge grounded conversation.
>
> Nevertheless, following your suggestion, we have conducted a refining experiment of LLM (GPT-4; > 1 trillion parameters) responses. We first generated the KGC response $\hat{u}$ with LLM. Then, we made LLM mine the key entities from the knowledge source and refine the given utterance. The prompts used in this experiment are the same as in Appendix H.
>
> As a result, all samples in the test set are refined with $\tau$=0.5. In the FoCus dataset, the refined utterances (marked with '+' in Table) show about a 2-3% increment in EC and TC and a 5-7% increment in DAE. Also, in WoW and CMUDoG, the result shows a similar tendency with the experiments in the paper, and our refining method helps improve the source-faithfulness scores (e.g., EC, TC and DAE)as shown below.
>
>
>
> |Data|Model|EC|TC|DAE|chrF|R-L|
> |--- |--- |--- |--- |--- |--- |--- |
> |FoCus|LLM_KG|42.74|26.72|0.84|38.16|40.64|
> ||+LLM|44.95|27.45|0.89|39.61|42.20|
> ||+LLM_REM|**45.83**|**28.46**|**0.91**|**40.01**|**41.98**|
> |WoW|LLM_KGC|6.91|5.87|0.36|18.68|18.65|
> ||+LLM|9.27|6.73|0.47|21.24|20.17|
> ||+LLM_REM|**13.28**|**9.25**|**0.63**|**25.17**|**21.19**|
> |CMUDoG|LLM_KGC|4.77|7.24|0.29|**14.36**|**13.44**|
> ||+LLM|5.61|8.53|0.38|**14.36**|12.21|
> ||+LLM_REM|**7.88**|**14.26**|**0.54**|**14.36**|10.04|
>
> This experiment confirms that LLM utterance suffers from entity hallucination in generating answers as in [1] and it shows the need for post-hoc refining. This result will be further included in Appendix of the paepr.
>
> &nbsp;
>
>     Questions For The Authors: For the LLM experiment (Section 4.6), are the LLMs just used to refine the BART responses?
>
> Re: Yes, LLM was used to refine the BART responses in Section 4.6. Also, in Section 4.3, we show the case that BART was used to refine the adversarial utterances of LLM.
>
> &nbsp;
>
>     Questions For The Authors: What happens when the LLMs generate their own responses from scratch, and REM is used to refine LLM responses? E.g. it would be interesting (but may not be the case) if REM (BART-large, 400M parameters) is still able to improve the responses of a larger model.
>
> Re: While the performance of a generative model is largely affected by its size, in terms of source-faithfulness, we believe that even smaller models can contribute because they are used only when refining the *erroneous* utterances of larger models, not refining all cases. However, as you mentioned, it would be interesting to see how the smaller model improves the responses of the larger models. Along with the experiments of LLM-to-LLM refining above, we plan to conduct and report additional experiments where the smaller model refines the larger models’ response to understand the performance and limitations of our method when we are given an extra page. We thank you for the insights and for helping us improve our paper.
>
> &nbsp;
>
> References
>
> [1] Ji, Z., Lee, N., Frieske, R., Yu, T., Su, D., Xu, Y., ... & Fung, P. (2023). Survey of hallucination in natural language generation. ACM Computing Surveys, 55(12), 1-38.Chicago

---

### Official Review · Reviewer_uiSe · 2023-08-04

**Soundness:** 4

**Excitement:**

3: Ambivalent: It has merits (e.g., it reports state-of-the-art results, the idea is nice), but there are key weaknesses (e.g., it describes incremental work), and it can significantly benefit from another round of revision. However, I won't object to accepting it if my co-reviewers champion it.

**Missing References:**

It will be helpful to discuss recent work focusing on knowledge-grounding, including the following, in relation to the proposed work:

Kim, S., Kwon, O. W., & Kim, H. (2020). Knowledge-grounded chatbot based on dual Wasserstein generative adversarial networks with effective attention mechanisms. Applied Sciences, 10(9), 3335.

Zhang, Y., Fu, H., Fu, C., Yu, H., Li, Y., & Nguyen, C. T. (2023, June). Coarse-To-Fine Knowledge Selection for Document Grounded Dialogs. In ICASSP 2023-2023 IEEE International Conference on Acoustics, Speech and Signal Processing (ICASSP) (pp. 1-5). IEEE.

Wang, W., Gao, W., Feng, S., Chen, L., & Wang, D. (2021, October). Adaptive posterior knowledge selection for improving knowledge-grounded dialogue generation. In Proceedings of the 30th ACM International Conference on Information & Knowledge Management (pp. 1989-1998).

Wu, T. W., & Juang, B. H. (2022, May). Knowledge Augmented Bert Mutual Network in Multi-Turn Spoken Dialogues. In ICASSP 2022-2022 IEEE International Conference on Acoustics, Speech and Signal Processing (ICASSP) (pp. 7487-7491). IEEE.

Grassi, L., Recchiuto, C. T., & Sgorbissa, A. (2022). Knowledge-grounded dialogue flow management for social robots and conversational agents. International Journal of Social Robotics, 14(5), 1273-1293.

**Paper Topic And Main Contributions:**

The work aims to address the challenge of hallucinations in language generation systems that are not faithful to the source information through an entity refinement method applied to generated utterances. The proposed system, Refining Method by Entitiy Mining (REM), seeks to enhance the quality of generated utterances. REM achieves this by leveraging source knowledge, i.e., mining key entities from the original source and implicitly using these mined entities to refine utterances, aiming to increase faithfulness to the original source of information.

The first step is identifying the utterances for refinement, done via a scoring method, DEA, which generates a score between 0 and 1 between the utterance and knowledge source. The candidates for refinement are filtered based on a score threshold, which is manually tuned and set to 0.5 for most settings for the paper. Post the filtering process, REM consists of two modules, an entity miner and an utterance generator, both trained in a multi-task setting with a joint training function.

The paper presents results on FoCus, WoW, and CMUDoG datasets. Four knowledge-grounded conversation (KGC) baselines are used, BART, EDMem, ITDD, and INFO, presented compared to REM variants. Results are shown for cross-data refinement, adversarial refining, and ablation studies.

**Questions For The Authors:**

The methodology for REM-LLM is not very clear, it would benefit from adding implementation details. Will the plug-and-lay system be shared along with trained models?

**Reasons To Accept:**

The paper presents a model to improve source faithfulness of generated utterances using the knowledge source, which can be used on top of LLMs.

The results show an improvement from using REM over baselines, the paper builds a strong case using multiple evaluation metrics.

The work also explains the reason for performance differences, including dataset compositions.

The paper lays out limitations well.

The models will be shared to reproduce results.

The work considers the carbon footprint of the training.

**Reasons To Reject:**

Human evaluation is done for a few samples per dataset and can be improved.

Important sections are in the appendix, particularly the literature review, which situates this work.

**Reproducibility:**

4: Could mostly reproduce the results, but there may be some variation because of sample variance or minor variations in their interpretation of the protocol or method.

**Reviewer Confidence:**

4: Quite sure. I tried to check the important points carefully. It's unlikely, though conceivable, that I missed something that should affect my ratings.

**Typos Grammar Style And Presentation Improvements:**

General comments:

Sections of the paper can be hard to parse, particularly sections 2, 3, and 4.

Acronyms are introduced without the full form, e.g., DEA, PLM, etc.

Key sections on related work and implementation are in the appendix.


Typos:

038: a large number of training corpora

122: incomplete sentence

162-164: typo in the value of tau

187-189: suggest rewriting the sentence for clarity

205: missing punctuation or incomplete sentence

211: consider omitting either both or two

263: add space between parentheses

275: produce fluent “text”

379: table 2 (not 9)

---

> ### Author Rebuttal · Authors · 2023-08-25
>
> We deeply thank the reviewer for their _insightful comment_ and detailed feedback on _missing references_, _typos_, and _presentation improvements_. They are exceedingly helpful for us to improve our paper. Please find our responses in the following.
>
> &nbsp;
>
>     Reasons To Reject: Human evaluation is done for a few samples per dataset and can be improved.
>
> Re: While it would be most objective to perform a human evaluation on all test data samples, given our financial situation, it was our best to hire three annotators to evaluate 100 samples. In particular, we believe that a sample of 300 completely randomly selected test data, without any manipulation, is sufficiently representative of our overall experimental results, given other studies that have conducted human evaluations in the dialogue domain and model generation [1, 2].
>
> &nbsp;
>
>     Reasons To Reject: Important sections are in the appendix, particularly the literature review, which situates this work.
>
> Re: Due to the page limit, we had to put the more important sections (e.g., Proposed Methods, Experiments) on the main pages, but we will put a literature review on the main pages for the completeness of the manuscript following your comments, when we are given one extra page.
>
> &nbsp;
>
>     Questions For The Authors: The methodology for REM-LLM is not very clear, it would benefit from adding implementation details.
>
> Re: We attached the prompts used for REM-LLM in Appendix H, but we will add more detailed implementations regarding the information of the prompts and the models.
>
> &nbsp;
>
>     Questions For The Authors: Will the plug-and-lay system be shared along with trained models?
>
> Re: Our plug-and-play method can be used to refine utterances generated by any other models that have already been trained, or it can be trained with existing KGC models from scratch by jointly minimizing the training loss (KGC loss and REM loss).
>
> &nbsp;
>
> References
>
> [1] Li, Y., Peng, B., Shen, Y., Mao, Y., Liden, L., Yu, Z., & Gao, J. (2021). Knowledge-grounded dialogue generation with a unified knowledge representation. *arXiv preprint arXiv:2112.07924*.
>
> [2] Adolphs, L., Shuster, K., Urbanek, J., Szlam, A., & Weston, J. (2021). Reason first, then respond: Modular generation for knowledge-infused dialogue. *arXiv preprint arXiv:2111.05204*.

---

### Official Review · Reviewer_fxJP · 2023-08-04

**Soundness:** 4

**Excitement:**

4: Strong: This paper deepens the understanding of some phenomenon or lowers the barriers to an existing research direction.

**Paper Topic And Main Contributions:**

The authors implement and test a refinement model for utterances generated by Knowledge grounded conversation (KGC) models.
IN KGC, hallucination, especially entity-level hallucination, is a major issue that makes conversation fail completely. The authors design a REM model for refining non-truthful utterances produced by some KGC model using a bipartite training strategy, making use of a NER head and a language generation head, both trained with a separate log likelihood loss. The overall loss being the sum of the two. They define their refinement model probabilty P(u_refine|k, û) as proportional to the product P(e|k,û) P(u_refine|e,k,û).

So, the REM model uses a mining module to extract information, takes the non-optimal output of the KGC model, scores this for unfaithfulness against the knowledge source and then conditions on both in order to produce the revised utterance.

Numerous different metrics were used for evaluation, specialized for knowledge truthfulness, on numerous corpora.
REM decreases diversity metrics, since refinement removes hallucinations that may increase those metrics.
BUT REM strongly improves source-faithfulness!
The ablation study shows that without the entity miner, performance decreases (though still better as baseline), therefore entity mining is of essential importance for REM performance and source faithfulness of KGC.

Human evaluation is performed as well, showing that in majority, the refined utterances are rated best. Concepts of evaluative criteria are opaque, but this is an inherent problem of metrics for human evaluation and not an issue of this paper.





**Questions For The Authors:**


A question that was haunting me throughout the paper is whether we would see a comparable increase in truthfulness to the knowledge source if the REM model was directly implemented as KGC model instead of a refinement model that is used for post-hoc improvement on top of the actual KGC model.

Furthermore, what influence does the quality of the utterance to be refined (û) have on the REM output quality? is there a relation?

**Reasons To Accept:**

They take an interesting approach to refine utterances that were already produced, making the system quite versatile. The performance improvement in regard to knowledge source truthfulness is good. The paper is well written and mostly easy to understand.

**Reasons To Reject:**

The choice of putting the related work and the implementational detail into the appendix is irritating.

**Reproducibility:**

4: Could mostly reproduce the results, but there may be some variation because of sample variance or minor variations in their interpretation of the protocol or method.

**Reviewer Confidence:**

4: Quite sure. I tried to check the important points carefully. It's unlikely, though conceivable, that I missed something that should affect my ratings.

**Typos Grammar Style And Presentation Improvements:**

p. 2 line 122: incomplete sentence, I think one of the authors accidentially deleted a sentence segment here.

---

> ### Author Rebuttal · Authors · 2023-08-25
>
> Thanks for your constructive comments and suggestions. We sincerely appreciate your time reading the paper, and our responses to your comments are below.
>
> &nbsp;
>
>     Reasons To Reject: The choice of putting the related work and the implementational detail into the appendix is irritating.
>
> Re: Due to the page limit, we had to put the more important sections (e.g., Proposed Methods, Experiments) on the main pages. However, we agree that the Related Work and Implementation Details will help improve the understanding of the readers. We will place it on the main pages with the extra page given if our manuscript is accepted.
>
> &nbsp;
>
>     Questions For The Authors: A question that was haunting me throughout the paper is whether we would see a comparable increase in truthfulness to the knowledge source if the REM model was directly implemented as KGC model instead of a refinement model that is used for post-hoc improvement on top of the actual KGC model.
>
> Re: In general, it is known that direct training helps to increase performance [1]. However, in this case, based on experimental results that show better performance with refinement only when needed than with refinement even when not needed, direct training may decrease performance and efficiency because it forces the model to refine every case. As a result, we aim to refine only the failed utterances, which are not source-faithful, in a post-hoc manner. Additionally, this method ultimately reduces the number of training parameters because it does not require additional training of existing models, and it can be applied to models with undisclosed parameters, such as ChatGPT, in a plug-and-play manner.
>
> &nbsp;
>
>     Questions For The Authors: Furthermore, what influence does the quality of the utterance to be refined (û) have on the REM output quality? is there a relation?
>
>
> Re: We observed that the length of the utterance to be refined $\hat{u}$ influences the refined output; inputs with shorter lengths tended to have shorter outputs. We will add this observation to improve our paper.
>
> &nbsp;
>
>     Typos Grammar Style And Presentation Improvements:
>
> Re: Regarding the typos, we appreciate you for letting us know. We will modify it and improve our manuscript.
>
> &nbsp;
>
> References
>
> [1] Ramos, A. G. C. P., Mehrotra, A., Lane, N. D., & Bhattacharya, S. (2021, October). Conditioning sequence-to-sequence networks with learned activations. In *International Conference on Learning Representations*.

---

### Meta-Review · Area_Chair_Ezey · 2023-09-25

**Recommendation:** 4

**Metareview:**

Interesting post-hoc refinement technique, potentially extendable to other knowledge-grounded generative tasks. It also contains important checks for source-faithfulness or provenance.

---

### Decision · Program_Chairs · 2023-10-07

**Decision:**

Accept-Main

**Comment:**

Interesting post-hoc refinement technique, potentially extendable to other knowledge-grounded generative tasks. It also contains important checks for source-faithfulness or provenance.